# PD-Refiner: An Underlying Surface Inheritance Refiner with Adaptive Edge-Aware Supervision for Point Cloud Denoising

Chengwei Zhang*
National Key Laboratory of
Microwave Imaging, Aerospace
Information Research Institute,
Chinese Academy of Sciences
Beijing, China

Xueyi Zhang*
Laboratory for Big Data and Decision,
National University of Defense
Technology
Changsha, Hunan, China

Xianghu Yue
Department of Electrical and
Computer and Engineering, National
University of Singapore
Singapore, Singapore

Mingrui Lao
National Key Laboratory of
Information Systems Engineering,
National University of Defense
Technology
Changsha, Hunan, China

Tao Jiang
National Key Laboratory of
Microwave Imaging, Aerospace
Information Research Institute,
Chinese Academy of Sciences
Beijing, China

Jiawei Wang
School of Software Engineering,
Tongji University
Shanghai, China

Fubo Zhang
National Key Laboratory of
Microwave Imaging, Aerospace
Information Research Institute,
Chinese Academy of Sciences
Beijing, China

Longyong Chen†
National Key Laboratory of
Microwave Imaging, Aerospace
Information Research Institute,
Chinese Academy of Sciences
Beijing, China

## Abstract

Point clouds from real-world scenarios inevitably contain complex noise, significantly impairing the accuracy of downstream tasks. To tackle this challenge, cascading encoder-decoder architecture has become a conventional technical route to iterative denoise. However, circularly feeding the output of denoiser as its input again involves the re-extraction of underlying surface, leading to unstable denoising process and over-smoothed geometric details. To address these issues, we propose a novel denoising paradigm dubbed PD-Refiner that employs a single encoder to model the underlying surface. Then, we leverage several lightweight hierarchical Underlying Surface Inheritance Refiners (USIRs) to inherit and strengthen it, thereby avoiding the re-extraction from the intermediate point cloud. Furthermore, we design adaptive edge-aware supervision to improve the edge awareness of the USIRs, allowing for the adjustment of the denoising preferences from global structure to local details. The results demonstrate that our method not only achieves state-of-the-art performance in terms of denoising stability and efficacy, but also enhances edge clarity and point cloud uniformity.

*These authors contributed equally to this work.
†Corresponding author: chenly@aircas.ac.cn

## CCS Concepts

• **Computing methodologies** → **Point-based models**.

## Keywords

Point Cloud Denoising, Point Cloud Processing, Geometry Detail Recovery, Edge-Aware Supervision

**ACM Reference Format:**
Chengwei Zhang, Xueyi Zhang, Xianghu Yue, Mingrui Lao, Tao Jiang, Jiawei Wang, Fubo Zhang, and Longyong Chen. 2024. PD-Refiner: An Underlying Surface Inheritance Refiner with Adaptive Edge-Aware Supervision for Point Cloud Denoising. In *Proceedings of the 32nd ACM International Conference on Multimedia (MM '24), October 28-November 1, 2024, Melbourne, VIC, Australia.* ACM, New York, NY, USA, 10 pages. https://doi.org/10.1145/3664647.3681540

## 1 Introduction

As a fundamental representation of 3D geometry, point cloud can be acquired through various methods, including LiDAR [18, 24, 59], millimeter wave radar [36, 42, 44, 51], and synthetic aperture radar [37, 62, 63]. It plays a pivotal role in the realm of 3D vision, finding applications in diverse areas, such as object detection [7, 48, 49], classification [6, 16, 20], segmentation [22, 66, 68], and surface reconstruction [32, 33, 47]. It can also integrate with other modalities [15, 25, 60, 64, 70] and apply across various domains, including autonomous driving [4, 18] and urban modeling [12, 35, 54]. Nevertheless, owing to sensor limitations and environmental factors, collected point clouds often contain intricate noise, which will significantly impacts downstream tasks' accuracy. Consequently, the development of point cloud denoising algorithms to mitigate noise is a critical research area.

Current methods [5, 26, 28, 40] typically perform denoising using single or cascaded structurally identical denoisers in both the training or testing phases. As illustrated in Fig. 1(a), single-step denoiser like GPDNet [40] employs a pair of encoder and decoder for denoising. On the one hand, the encoder's role is to learn the relationship between each point and its neighbors in a noisy point cloud, thereby extracting the underlying surface representations. On the other hand, the decoder aims to regress the position displacement vector based on its representation to update point position. To further remove residual noise, iterative denoising methods (*e.g.*, Score [28], PSR [2], IPFN [5]), as in Fig. 1(b), involve iterating single-step methods during testing or training phases. The intermediate point cloud with residual noise from the previous denoiser will serve as the input to the next denoiser for further denoising.

Although previous methods remove point cloud noise to some extent, there are certain limitations in these methods, obstructing the further improvements of denoising capabilities. **Firstly, re-extracting representations from intermediate point clouds may destabilize the denoising process.** In the high noise and low density point clouds, it is challenging to accurately estimate the underlying surfaces with a single denoiser, resulting in inevitable residual noise and edge over-smoothness. Subsequent iterations intensify these issues, as the re-extracted underlying surfaces lead to points converging towards a biased surface, ultimately causing an unstable denoising process and loss of geometric detail. **Secondly, existing methods typically lack accurate and adaptive supervision that adjusts the intensity of edge awareness according to the denoising stages.** In the initial denoising process, it is common for denoisers to prioritize non-edge regions due to their simpler geometry and lower noise sensitivity, while edge regions are challenging to restore with limited prior knowledge. Then, noise reduction in non-edge areas makes edge regions more discernible, enabling the recovery of their geometric details. However, without adaptive intensity of edge awareness, there might be an inappropriate focus on the difficult-to-recover edge regions in the early stages. Such undesired behaviour hampers the network's stability and convergence, making it less effective at accurately preserving geometric details during the denoising process.

This paper introduces PD-Refiner, as in Fig. 1(c), to address the aforementioned limitations of previous methods. PD-Refiner aims to improve denoising stability by inheriting and strengthening the underlying surface representation from the prior denoising step, thus avoiding the re-extraction of representations from the intermediate point cloud. Specifically, PD-Refiner uses a single encoder to extract representation from the original noisy point cloud. Then, it employs hierarchical USIRs to inherit and strengthen the point-wise representation by synchronously considering the moving trend of the point and the multi-scale neighborhood modeling. This approach ensures that USIRs at different stages utilize a stable and increasingly refined underlying surface estimation for updating point positions. Consequently, this stable denoising process alleviates the denoisers' learning burden of aligning points to the underlying surface, which, in turn, allows more learning capacity to be devoted to improving the quality of the point cloud, such as enhancing uniformity and restoring edges. Additionally, to more precisely guide the USIRs in edge restoration, we design a novel adaptive edge-aware supervision strategy named Adaptive Shape

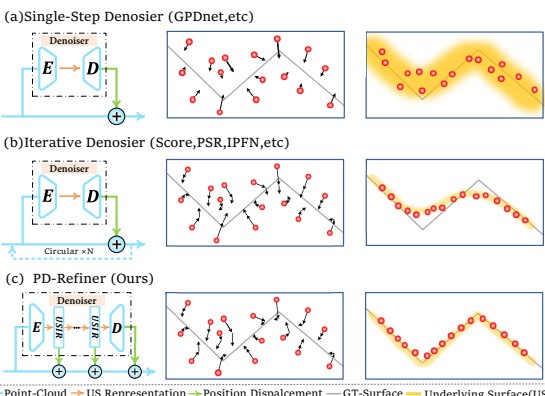

**Figure 1: Comparisons of PD-Refiner with previous single-step and iterative denoising methods.**

Preserving Loss (ASPL) to gradually recover geometric details. ASPL adjusts the intensity of edge awareness within the optimization targets of each USIR layer based on the denoising step number. It guides the shallow USIRs to prioritize denoising the global non-edge point (plane) and gradually provides stronger supervision signals to the edge points (local geometric detail) in deeper USIRs. Feature extraction is performed only once in the entire denoising process. Additionally, benefiting from the single-layer point convolution, the USIR architecture is more lightweight compared to traditional encoders, thereby requiring fewer computing resources than conventional stacked encoder-decoder denoising methods. The main contributions of this paper are:

- We propose PD-Refiner, a novel yet practical point cloud denoising framework that boosts denoising stability by inheriting and strengthening the underlying surface representation without re-extracting it from intermediate point clouds. This methodology ensures a more stable denoising process by maintaining a consistent representation across denoising steps.

- We establish a stable denoising process that promotes the network to focus more on restoring geometric details. To this end, we introduce the Adaptive Shape Preserving Loss (ASPL), an adaptive supervision strategy that precisely adjusts the edge-awareness intensity of each USIR layer, ensuring detailed and accurate edge restoration.

- Extensive experiments demonstrate that PD-Refiner achieves superior performance compared with previous SOTA iterative methods in both testing or training phases with demanding fewer computational resources. It is noteworthy that, it also benefits the promoting of denoising quality, including the local geometric details and the uniformity.

## 2 Related Work

Deep learning driven point cloud denoising methods have demonstrated remarkable denoising capabilities in various noise modalities [8–11, 14, 21, 23, 34, 65]. The rapid progress of point cloud analysis has led to the development of various neural network architectures and operators tailored to processing point cloud data such as PointNet [43], DGCNN [56], PointTransformer [67], etc. Deep learning driven point cloud denoising methods can be broadly categorized into displacement-based and generation-based approaches.

**Displacement-based methods** like GPDNet [40, 41] utilizes graph-convolutional layers [50] as encoder to elegantly addresses the permutation invariance problem commonly encountered in learning-based point cloud denoising methods. Noise2noise [30] proposes to learn Signed Distance Functions (SDF) [29] through noise-to-noise mapping and update the point position.

Iterative denoising plays a crucial role in high noise point cloud. Conventional iterative denoising methods train single denoiser and conduct testing phase multiple times. PathNet [57] designs dynamic network that adjust the denoising path according to geometry and noise level of each point. RePCD [3] proposes to use recurrent neural networks (RNNs) to model the iterative denoising process. IPFN [5] achieves iterative denoising by stacking multiple IterationModules during training. These methods [3, 5] directly add the position displacement to the coordinates of the point to denoise. There are also methods to introduce step size to control the movement of points in iterative denoising. They model the gradients of point cloud distribution to reduce noise through gradient ascent. Score [28] introduces a gradient ascent method to map 3D distribution to a 2D manifold and utilize Langevin sampling [1, 27] to iteratively recover low-noise data from high-noise input. PSR [2] introduced a gradient-based point set resampling that continuously model the probability distribution density of the noisy point cloud, and resample this distribution to obtain the denoised point cloud. MAG [69] introduced a momentum term in the gradient update process to deal with fluctuate gradient field. *Although iterative denoising has achieved a certain effect, these methods generate the intermediate point cloud and need the re-extraction of underlying surfaces, causing the unstable denoising process. Our method falls into displacement-based method, but unlike the previous approach, we model the representation of the point cloud for once, instead of re-extracting from the output point cloud from the previous step.*

**Generation-based methods** focus on removing high-noise points and generating low-noise points. DMR [26] estimates the point-wise noise level as a measure of point-wise quality. They select the part with the highest quality and generate new points to restore the original point count. Random screen [55] proposes integrating points into higher-dimensional sub-patches to reduce the number of points and denoise, then decoupling them to recover the original number of points. SSPCN [19] suggests extracting features from the down-sampled points for feature compensation, alleviating the inconsistency between point features and their actual locations. *While this method has advantages in dealing with outliers, the downsampling process results in the loss of geometric details. On the contrary, our method could maintain more detailed geometric information without the points sampling.*

## 3 Methods

### 3.1 Preliminary

Following previous state-of-the-art methods [5, 26, 28], we start with a clean point cloud $PCD_{clean} \in \mathbb{R}^{M \times 3}$, which is normalized to the unit sphere and $M$ is the point number of the object. An additive 3D Gaussian noise $\xi \sim N(0, \sigma)$ is added to the clean point cloud to get a noisy point cloud $PCD_{noisy} \in \mathbb{R}^{M \times 3}$, where $PCD_{noisy} = PCD_{clean} + \xi$. The noise intensity is controlled by the variance $\sigma$. To create paired training data, we randomly select a reference point

$x_r$ from $PCD_{noisy}$, and choose the nearest $N$ points, where $N$ is the patch size, from $PCD_{noisy}$ and $PCD_{clean}$ to get $P_n$ as the input noisy point cloud and $P_c$ as the ground truth clean point cloud.

Displacement-based point cloud denoising networks commonly utilize an encoder, to extract point-wise underlying surface representations based on the noisy points 3D coordinates. This process can be formulated as:

$$F_n = Encoder(P_n, \theta^E), \tag{1}$$

where $F_n \in \mathbb{R}^{N \times C}$ is the extracted point-wise underlying surface representation and $C$ is the feature dimension. The encoder is built upon Dynamic EdgeConv layers [5, 50]. The update of point-wise features is formulated as:

$$h_i^{s+1} = (f_\phi(h_i^s) || \sum_{j:(i,j) \in \epsilon} g_\theta(h_i^s || h_j^s - h_i^s)), \tag{2}$$

where $h_i^s$ and $h_j^s$ represent the feature of the center point and its neighborhood points in layer $s$, $\epsilon$ is the edges, $f_\phi(\cdot)$ and $g_\theta(\cdot)$ are parameterized by MLPs, and $(\cdot || \cdot)$ denotes concatenate.

The $i$-th point-wise representation encodes information of the underlying surface formed by the neighborhood of $i$-th points and the relationship between $i$-th point and the underlying surface. A decoder, composed of MLPs, is utilized to map the relationship in the representation into position displacement vector, updating the coordinates accordingly to bring the point closer to the underlying surface. This process can be formulated as:

$$P^{disp} = Decoder(F_n, \theta^D), \tag{3}$$

where $P^{disp} \in \mathbb{R}^{N \times 3}$ is the predicted position displacement vector. The final denoised point cloud is obtained as $P_d = P_n + P^{disp}$. The training objective of the denoising network is to minimize the Chamfer Distance (CD) between the denoised point cloud $P_d$ and the clean point cloud $P_c$.

Iterative denoising is a commonly-used process to improve denoising performance, particularly for high-noise point clouds. It involves stacking multiple denoising networks formulated as:

$$F_l = Encoder(P_l^m, \theta_l^E)$$
$$P_{l+1}^m = P_l^m + Decoder(F_l, \theta_l^D), \tag{4}$$

where $\tau$ is the iteration number and $0 \leq l < \tau$, $P_l^m$ is the intermediate point clouds, $P_0^m = P_n$ representing the input noisy point cloud, and $P_d = P_\tau^m$ is the output denoised point cloud. $F_0 = F_n$ is the extracted original underlying surface representation from $P_n$. In testing phase iterative denoising [2, 28, 69], they use the same network where $\theta_i^u = \theta_j^u, u \in \{E, D\}, 0 <= i, j < \tau$. However, taking the intermediate point cloud with different residual noise as input of the same network increases the burden of learning point cloud distribution. To alleviate this problem, IPFN [5] propose iterative denoising in training phase to stage the iterative denoising process. Nevertheless, these iterative denoising methods need re-extracting representations from intermediate point clouds, resulting in the unstable denoising process. To address these limitations of iterative denoising, we introduce PD-Refiner as shown in Fig. 2, consisting of the Underlying Surface Inheritance Refiner (USIR), Shape Variance Weighting Network (SVWN) and adaptive edge-aware supervision strategy named Adaptive Shape Preserving Loss (ASPL).

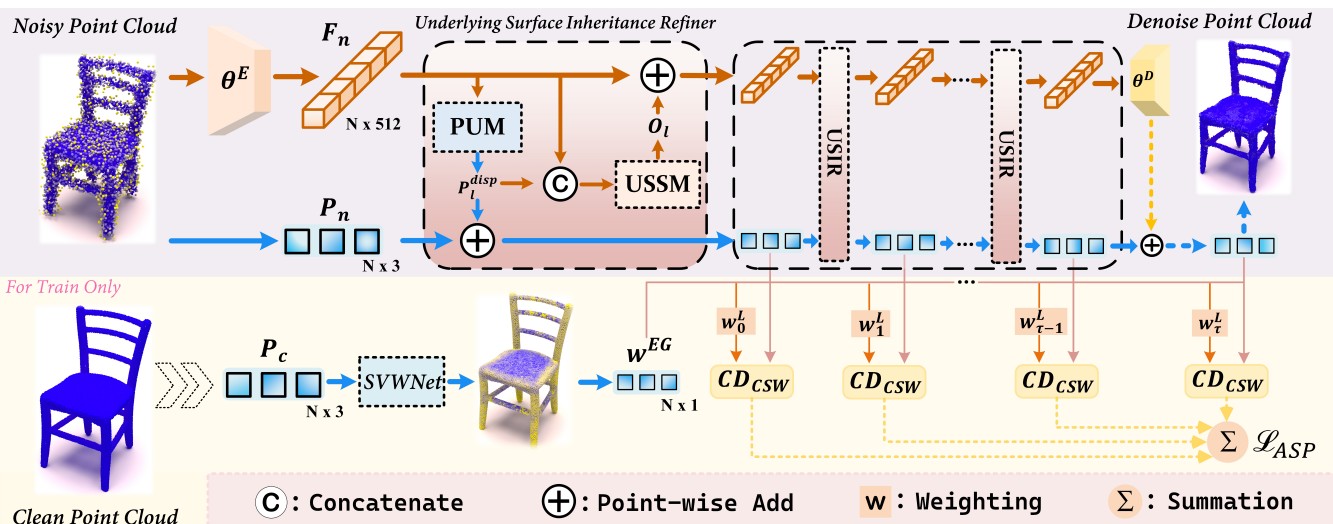

**Figure 2: Overview of the proposed PD-Refiner. Our PD-Refiner employs a single encoder along with a hierarchical Underlying Surface Inheritance Refiner (USIR) to strengthen the inherited underlying surface representation, ensuring a stable and accurate denoising process. We incorporate an edge-aware supervision strategy, called Adaptive Shape Preserving Loss (ASPL), to restore geometric details in the denoised point cloud.**

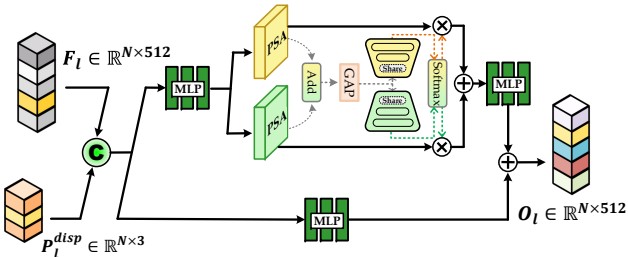

**Figure 3: Details of the USSM. The point-wise representation and position displacement vector is synchronously considering for multi-scale modeling.**

## 3.2 Underlying Surface Inheritance Refiner

Previous iterative denoising consists of a stack of encoder-decoder pairs, where each pair represents a denoiser. Unlike the network structure of previous iterative methods, PD-Refiner employs a single pair of encoder and decoder. We insert hierarchical USIRs between them to inherit and strengthen the underlying surface representation while update the point position.

First, we utilize a MLPs-based Position Update Module (PUM) to compute a positional displacement vector $P_l^{disp}$ using $F_l$ as input, enabling point position updates formed as $P_{l+1}^m = P_l^m + P_l^{disp}$, where $l$ signifies the layer number, and $F_l$ is the input representation of the $l$-th USIR. Then, we propose the Underlying Surface Strengthen Module (USSM), as shown in Fig. 3., that leveraging multi-scale neighborhood modeling to strengthen the estimation of the underlying surface. Benefited from the commonality of underlying surfaces described by neighboring points, this method combines multiple estimates and further achieves more robust and reliable results, thereby enhancing both the overall stability and accuracy of

the denoising process. Simultaneously, the positional displacement vector contains complementary information about point movement trend, also offering valuable guidance for estimating the underlying surface's location.

The input of USSM is formed as $G_l = (P_l^{disp} || F_l)$, where $G_l \in \mathbb{R}^{N \times (512+3)}$. Multi-scale neighborhood representation modeling is enabled by leveraging the selective kernel operation [17, 38, 39], specialized for dynamically selecting and aggregating convolutional kernels of various receptive field. The local geometry structure and noise distribution characteristics of point clouds present different scale requirements for underlying surface modeling. USSM adaptively fuses multi-scale underlying surface information learned from point moving trends and neighborhood representations to strengthen the underlying surface representation and formulated as $O_l = USSM(G_l)$ and $F_{l+1} = F_l + O_l$.

At the beginning of USSM, a MLP layer is used to adjust the feature dimension and get $G_l^a \in \mathbb{R}^{N \times 512}$. Multi-scale feature is extracted by Point Self-Attention (PSA) [52] layer to get multi-scale information formulated as:

$$H_l^a = \{PSA(G_l^a, k_1), ..., PSA(G_l^a, k_n)\}, \qquad (5)$$

where $k_o, 1 \le o \le n$ is the receptive fields. Considering the balance of computation and precision, we set $\{k_o\}_{o=1}^n = \{8, 16, 32\}$. These multi-scale features are added together and get the joint representation by global average pooling (GAP) through all points. Then, we utilize a MLP layer to squeeze the feature dimension to obtain the compressed mixed-scale feature $S_l^a \in \mathbb{R}^{N \times 128}$. The feature dimension is restored by multiple MLPs and use SoftMax to calculate the corresponding weights of each feature at different scale. We perform weighted summations of the multi-scale feature, and feed it into the MLP to get $O_l^a \in \mathbb{R}^{N \times 512}$. Residual connection is achieved by using an MLPs to get the residual feature $O_l^b \in \mathbb{R}^{N \times 512}$. The multi-scale feature and the residual feature is added as $O_l = O_l^a + O_l^b$.

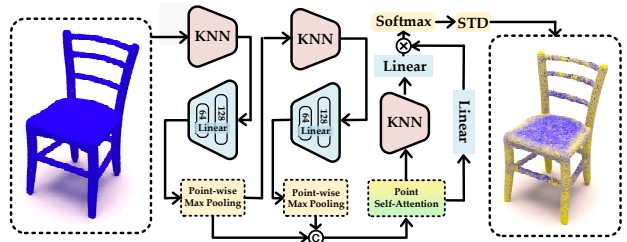

**Figure 4: Details of SVWN. High-weighted edge and low-weighted plane points are marked in yellow and blue.**

## 3.3 Shape Variance Weighting Network

Implementing ASPL requires knowing whether each point belongs to the edge region. We design a Shape Variance Weighting Network (SVWN), as in Fig. 4, and introduces attention-based edge point learning strategy [58] that leveraging statistical characteristics to extract edge weight from the clean point cloud patch $P_c$. SVWN calculates the variance of the attention-based correlation map within the neighborhood points feature. This calculation captures the degree of geometric structure change. It represents the probability that a point belongs to the edge region.

To be specific, SVWN construct dynamic graph structures [56] according to the feature similarity among points by KNN. MLP layers is used for point relationship learning. We use two dynamic graph convolution layers to encourage the network to organize the graph semantically and expands the receptive field of local neighborhood to better capture edge feature. The intermediate features from the pooling layer are concatenated and subsequently passed through a PSA layer to get the edge feature $F^{Edge} \in R^{N \times 128}$. KNN generates the edge feature difference $F^{Edge}_{diff}$ between the central point and its neighboring points. These features, $F^{Edge}$ and $F^{Edge}_{diff}$, respectively serve as the inputs for two MLPs, producing $F^{Query} \in R^{N \times 1 \times 128}$ and $F^{Key} \in R^{N \times 128 \times 32}$. The variance of attention-based correlation map is formed as $w^{Edge} = STD(Softmax(F^{Query} \times F^{Key}))$. A higher variance indicates a more pronounced change in the local geometry structure, signifying its probability of proximity to the edge. Since the edge of input patch does not belong to the real edge of the object, Gaussian weights is added to retain more accurate edge annotation. The final edge weight is formulated as:

$$w_i^{EG} = w_i^{Edge} * \frac{exp(-||x_i - x_r||_2^2/r_s^2)}{\sum_i exp(-||x_i - x_r||_2^2/r_s^2)}, \quad (6)$$

where $r_s = r/3$ and $r$ is the patch radius. The parameters of SVWN is updated through back propagation together with USIRs, without additional GT geometrical loss supervision like local curvatures.

## 3.4 Adaptive Shape Preserving Loss

To improve PD-Refiner's structural recovery, we propose Adaptive Shape Preserving Loss (ASPL), which adjusts the intensity of edge awareness within the optimization targets of each USIR layer based on the denoising step number to guide points precisely converge to the correct surface.

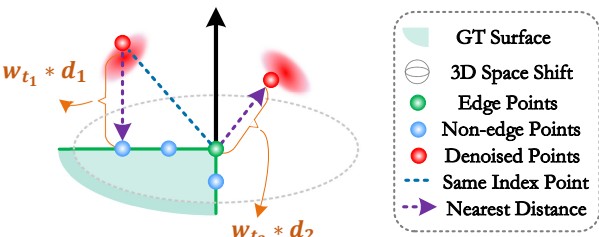

**Figure 5: Illustration of adding edge weights to different loss items. If add weights to both items $w_{t1}$ and $w_{t2}$, the supervision signal of the edge region might leak into the non-edge region.**

The Chamfer Distance (CD) loss is wildly used for supervised point cloud denoising, which can be formulated as:

$$CD(P_d, P_c) = \frac{1}{N}(\sum_{a \in P_d} min_{b \in P_c}||a - b||^2 + \sum_{b \in P_c} min_{a \in P_d}||a - b||^2), \quad (7)$$

where $||a - b||^2$ represents the Euclidean distance between the two points. Standard CD loss consists of two terms. The first term calculates the average nearest distance from each point in the denoised point cloud $P_d$ to the clean point cloud $P_c$, and the second term does the same for each point in $P_c$ to $P_d$. Within each term, every point's nearest distance contributes equally to the overall loss.

The edge weights extracted by SVWN can provide index-level edge weight annotation to $P_c$. Previous method [3] add weights to both terms (denoised-clean and clean-denoised) of CD loss according to point index. Considering the edge points in $P_c$ may shift to non-edge region due to added noise and denoising process, even though they have the same index, as shown in Fig. 5, the high weight to the edge region may leak to the non-edge region thus providing imprecise supervision signal of the edge region. To overcome this limitation, we propose an improved CD loss named Clean-denoised Shape Weighted CD formulated as:

$$CD_{csw}(P_d, P_c, w^L) = \frac{1}{N}(\sum_{a \in P_d} min_{b \in P_c}||a - b||^2$$
$$+ \sum_{b \in P_c} min_{a \in P_d}(1 + w_i^{EG} * w^L) * ||a - b||^2), \quad (8)$$

where only the loss term of the nearest distance from clean to denoised point cloud is added the edge weights. Since $P_c$ is not affected by noise addition or denoising, we can provide a more accurate supervision signal to edge region, and $w^L$ is used to control the intensity of edge awareness according to denoising stages.

To gradually increase the overall focus on the edge regions of USIR, we set different layer weights for the aggregation of losses at each layer. The layer weight is increasing with the number of USIR. The final ASPL is formulated as:

$$\mathcal{L}_{ASP} = \sum_{l=1}^{\tau} CD_{csw}(P_{m,l}, P_c, w_l^L), \quad (9)$$

where $w_l^L = \frac{exp(l)}{exp(\tau)}$ represents the layer weight.

| #Points | | 10K Points | | | | | | 50K Points | | | | | |
|---|---|---|---|---|---|---|---|---|---|---|---|---|---|
| Noise Level | | 1% Noise | | 2% Noise | | 3% Noise | | 1% Noise | | 2% Noise | | 3% Noise | |
| Method | Venue | CD↓ | P2M↓ | CD↓ | P2M↓ | CD↓ | P2M↓ | CD↓ | P2M↓ | CD↓ | P2M↓ | CD↓ | P2M↓ |
| PCN [45] | CGF 19 | 35.15 | 11.48 | 74.67 | 39.65 | 130.67 | 87.37 | 10.49 | 3.46 | 14.47 | 6.08 | 22.89 | 12.85 |
| GDPNet [40] | ECCV 20 | 37.80 | 13.37 | 80.07 | 44.26 | 134.82 | 91.14 | 19.13 | 10.37 | 50.21 | 37.36 | 97.05 | 79.98 |
| DMR [26] | ACM MM 20 | 44.82 | 17.22 | 49.82 | 21.15 | 58.92 | 28.46 | 11.62 | 4.69 | 15.66 | 8.00 | 24.32 | 15.28 |
| Score [28] | ICCV 21 | 25.21 | 4.63 | 36.86 | 10.74 | 47.08 | 19.42 | 7.16 | 1.50 | 12.88 | 5.66 | 19.28 | 10.41 |
| PD-Flow [31] | ECCV 22 | 21.26 | 3.81 | 32.46 | 10.10 | 44.47 | 19.99 | 6.51 | 1.64 | 11.73 | 5.81 | 19.14 | 12.10 |
| PSR [2] | TPAMI 22 | 23.53 | 3.06 | 33.50 | 7.34 | 40.75 | 12.42 | 6.49 | 0.76 | 9.97 | 2.96 | 13.44 | **5.31** |
| Noise2Noise [30] | ICML 23 | **10.60** | 2.41 | 29.25 | 10.10 | 42.21 | 18.47 | **3.77** | 1.55 | 10.29 | 4.84 | 16.54 | 9.72 |
| IPFN [5] | CVPR 23 | 20.56 | 2.18 | 30.43 | 5.55 | 42.41 | 13.76 | 6.05 | 0.59 | 8.03 | 1.82 | 19.71 | 10.12 |
| PathNet [57] | TPAMI 24 | 26.72 | 5.84 | 39.73 | 12.99 | 45.24 | 24.04 | 7.16 | 1.24 | 11.40 | 4.10 | 18.75 | 9.52 |
| **Ours**$_{tiny}$ | | 18.27 | 1.82 | 25.87 | 4.89 | 33.74 | 10.45 | 4.93 | 0.49 | 7.15 | 1.85 | 14.58 | 7.22 |
| **Ours** | | 17.54 | **1.66** | **24.44** | **4.49** | **30.77** | **9.13** | 4.66 | **0.45** | **6.53** | **1.64** | **12.28** | 5.75 |

**Table 1: Denoising comparison with SOTA methods in PU-Net dataset under Gaussian noise. CD is multiplied by $10^5$ and P2M is multiplied with $10^5$. Data in bold and underline represent the best and second best result among all methods.**

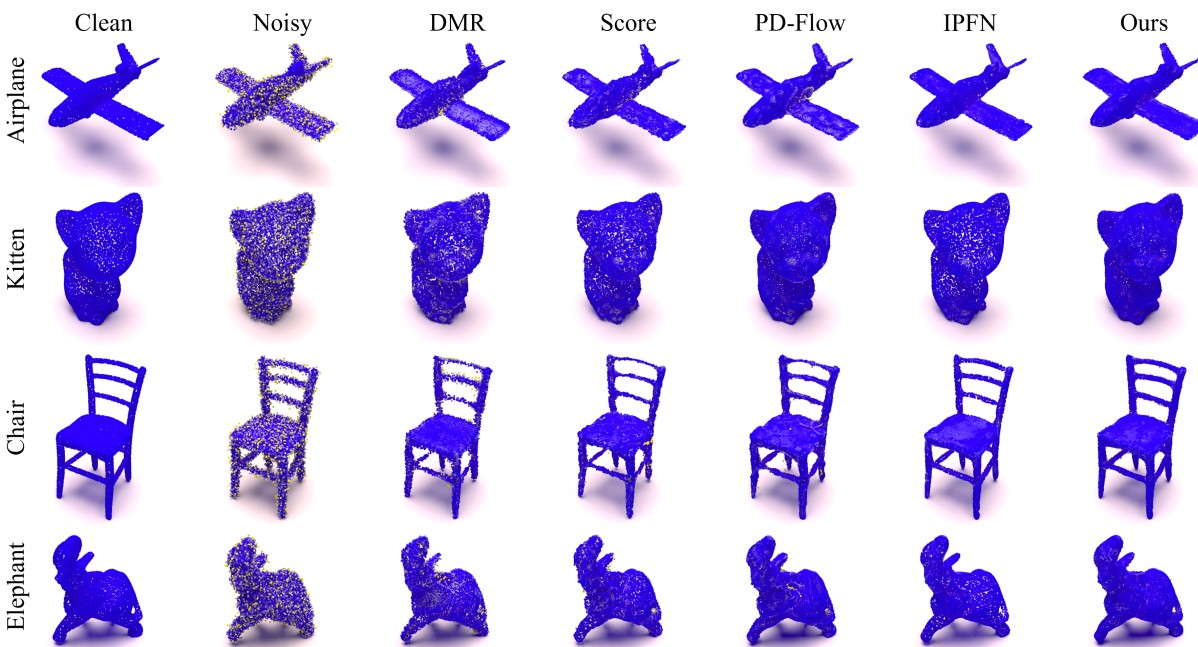

**Figure 6: Visual result of point-wise P2M distance for 10K resolution with Gaussian noise of 2% of the bounding sphere radius.**

## 4 Experiment

### 4.1 Dataset and Implementation Details

**Dataset:** We follow the training strategy of the previous methods that selects 40 meshes from the training set of PU-Net [61] for the network training. The training sample uses Poisson disk sampling with a resolution of 10k, 30k, 50k to obtain the point cloud, resulting in a total of 120 different training samples. We add Gaussian noise of 0.5% to 2% of bounding sphere's radius variance for each sample. The patch size $N$ of the training sample pair is set to 1000. The test set consist of 20 samples from the PU-Net test set with resolutions of 10k and 50k and noise intensity of 1%, 2% and 3%. CD and P2M are used to measure all method performance.

Furthermore, we conduct qualitative study on the real-world dataset Paris-Rue-Madame [46], which comprises scans obtained

through the utilization of the Mobile Laser Scanning system L3D2. Notably, this dataset exhibits real-world imperfections introduced by the inherit limitations of the scanning technology, making it an ideal benchmark for assessing the algorithm's robustness in the face of such artifacts. Next, we consider the Kinect v1 and Kinect v2 datasets [53], consisting of 71 and 72 real-world scans acquired using Microsoft Kinect v1 and Kinect v2 cameras.

**Implementation details:** We use the network structure PyTorch implementation of IPFN [5] and set the IterationModule number to 1 as our baseline. The proposed PD-Refiner is trained and tested on NVIDIA 3090 GPUs using PyTorch 1.9.0 with CUDA 11.1. We train the network for 100 epochs, with the Adam [13] optimizer. We used the cosine learning rate scheduler to smoothly decrease the learning rate from an initial value of $1 \times 10^{-4}$ to a final value of $1 \times 10^{-7}$. Batch size is set to 16.

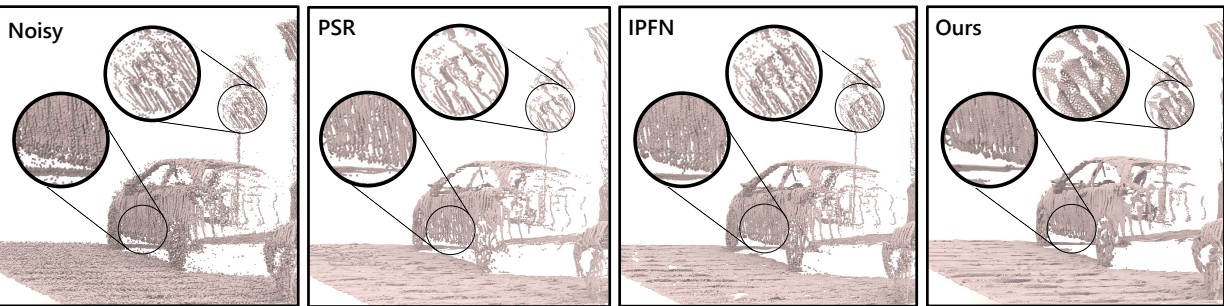

**Figure 7: Visual result of the real-world RueMadame dataset, showcasing the superior point uniformity and shape preservation ability of our method.**

| #Points | | 10K Points | | | | | |
|---|---|---|---|---|---|---|---|
| **Noise** | | **1% noise** | | **2% noise** | | **3% noise** | |
| **Type** | **Method** | **CD↓** | **P2M↓** | **CD↓** | **P2M↓** | **CD↓** | **P2M↓** |
| | PCN | 46.16 | 19.40 | 110.82 | 72.18 | 209.81 | 159.22 |
| | Score | 29.15 | 6.74 | 46.01 | 17.99 | 63.32 | 32.71 |
| **Lap** | PSR | 26.63 | 4.50 | 37.90 | 10.67 | 51.10 | 20.17 |
| | IPFN | 24.25 | 3.28 | 34.66 | 8.68 | 68.60 | 34.60 |
| | **Ours** | **20.62** | **2.66** | **28.96** | **7.94** | **42.48** | **17.93** |
| | PCN | 11.77 | 3.07 | 28.70 | 8.71 | 40.28 | 16.74 |
| | Score | 12.49 | 2.51 | 21.77 | 4.16 | 26.53 | 6.53 |
| **Dis** | PSR | 10.21 | 1.63 | 19.21 | 2.68 | 22.74 | 4.31 |
| | IPFN | 6.76 | 0.91 | 16.60 | 1.88 | 20.29 | 3.21 |
| | **Ours** | **6.13** | **0.75** | **15.34** | **1.50** | **19.49** | **2.56** |
| | PCN | 34.32 | 11.29 | 73.93 | 39.40 | 129.52 | 86.54 |
| | Score | 24.70 | 4.56 | 36.82 | 10.84 | 47.76 | 20.00 |
| **Ani** | PSR | 23.05 | 3.08 | 33.45 | 7.58 | 41.52 | 13.50 |
| | IPFN | 20.25 | 2.24 | 30.59 | 5.89 | 50.31 | 20.14 |
| | **Ours** | **17.41** | **1.76** | **24.70** | **4.80** | **33.43** | **11.28** |
| | PCN | 12.05 | 3.37 | 33.78 | 10.18 | 50.44 | 19.95 |
| | Score | 12.77 | 2.48 | 24.67 | 4.18 | 30.79 | 6.54 |
| **Uni** | PSR | 10.56 | 1.64 | 23.48 | 2.75 | 29.16 | 4.43 |
| | IPFN | 6.78 | 0.91 | 20.38 | 2.03 | 26.90 | 3.53 |
| | **Ours** | **6.12** | **0.76** | **17.11** | **1.52** | **21.65** | **2.62** |

**Table 2: Denoising comparison with SOTA methods in other noise distribution.**

## 4.2 Comparisons of the state-of-the-art

Tab. 1. shows the evaluation metrics comparison including CD and P2M under different noise level and point density. We run PathNet in PU-Net for 1x, 2x, and 4x testing phase denoising, under 1%, 2%, and 3% noise. Other methods were tested with default settings. We also offer a tiny version with $C = 256$ and $\tau = 4$. Our method significantly outperforms competitive methods in most settings. In particular, our method has proven to be more efficient, especially when encountering more complex, low-density point clouds with high noise levels, where the noise level far exceeds that encountered during training. This robustly demonstrates that PD-Refiner ensures a more stable denoising process and enhanced robustness.

The comparison of the visual results of the denoised point cloud from PU-Net obtained by the SOTA methods and PD-Refiner is shown in Fig. 6. The first two columns present the clean and noisy point clouds, serving as benchmarks for evaluating the effectiveness of various denoising methods. DMR shows significant loss

| Method | V1 | | V2 | |
|---|---|---|---|---|
| | **CD↓** | **P2M↓** | **CD↓** | **P2M↓** |
| **Noisy** | 15.36 | 8.21 | 25.21 | 14.60 |
| **PSR** | 15.18 | 7.77 | 23.25 | 12.74 |
| **PD-Flow** | **14.25** | 7.68 | 22.56 | 12.93 |
| **Score** | **14.25** | **6.94** | 23.46 | **12.36** |
| **IPFN** | 14.92 | 7.99 | 22.27 | 12.89 |
| **Ours** | 14.34 | 7.61 | **22.26** | 12.89 |

**Table 3: Denoising comparison with SOTA methods in Kinect V1 and V2 dataset.**

of geometric details and fails to remove residual noise effectively. Score performs better in noise reduction but struggles with surface smoothness. PD-Flow ensures a more uniform point distribution but suffers from irregular surface undulations and misalignment. IPFN achieves commendable noise reduction but lacks uniformity in point cloud distribution. Compared with previous methods, PD-Refiner excels in both residual noise elimination and preservation of geometric details. This superiority is attributed to an innovative underlying surface inheritance and strengthening framework, coupled with an adaptive shape preserving loss. This denoising process not only stabilizes but also improves the structural integrity and uniformity of the point cloud distribution.

Furthermore, we also test under various noise modalities including anisotropic Gaussian, discrete, Laplace and uniform noise. As in Tab. 2, PD-Refiner can achieve the state-of-the-art performance in all noise modalities and noise levels. In the real-world RueMadame dataset, as in Fig. 7, our method achieves better preservation of geometric details and point uniformity. The quantitative comparison of Kinect V1 and V2 dataset, as in Tab. 3, also demonstrate the generalizability of PD-Refiner in real-world scenarios.

## 4.3 Ablation study

We conduct an ablation study to verify the effectiveness of each component, including: 1) the influence of the number of USIRs used in PD-Refiner; 2) the inputs and methods for strengthening the underlying surface representation in USIR; 3) stage adaptability and edge supervision strategies in ASPL; and 4) the computational resource requirements across different configurations.

*The influence of the number of USIR used on PD-Refiner is studied in Tab. 4.* We incorporate different numbers of USIRs into the baseline model and conduct multiple testing phase denoising. The results of 1× testing phase denoising indicate that adding more

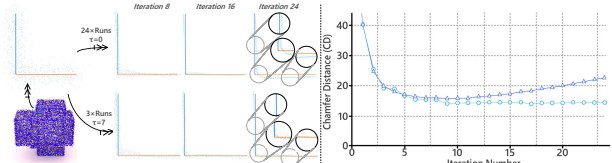

**Figure 8: Details of SVWN. High-weighted edge and low-weighted plane points are marked in yellow and blue.**

| #Points | | | 10K Points | | | | | |
|---|---|---|---|---|---|---|---|---|
| Noise Level | | | 1% noise | | 2% noise | | 3% noise | |
| Runs | $\tau$ | | CD↓ | P2M↓ | CD↓ | P2M↓ | CD↓ | P2M↓ |
| | 0 | | 19.91 | 2.68 | 29.05 | 6.60 | 43.74 | 17.30 |
| | 1 | | 18.70 | 1.94 | 27.02 | 5.37 | 36.07 | 11.69 |
| 1× | 3 | | 18.00 | 1.73 | 25.57 | 4.77 | 32.84 | 9.86 |
| | 7 | | 17.54 | 1.66 | 24.44 | 4.49 | 30.77 | 9.13 |
| 2× | 3 | | 17.72 | 1.70 | 24.51 | 4.54 | 30.74 | 9.33 |
| | 7 | | 17.53 | 1.64 | 23.92 | 4.34 | 29.77 | 8.89 |

**Table 4: Study of the influence of the number of USIR used on PD-Refiner. $\tau$ is the USIR number and we run multiple testing phase denoising.**

| #Points | | | | 10K Points | | | | | |
|---|---|---|---|---|---|---|---|---|---|
| Noise Level | | | | 1% noise | | 2% noise | | 3% noise | |
| $F_l$ | $P_l^{disp}$ | USSM | | CD↓ | P2M↓ | CD↓ | P2M↓ | CD↓ | P2M↓ |
| ✓ | ✓ | | | 19.39 | 2.39 | 28.37 | 6.23 | 43.23 | 16.95 |
| | ✓ | ✓ | | 18.56 | 1.89 | 26.77 | 5.12 | 35.56 | 11.54 |
| ✓ | | ✓ | | 17.86 | 1.70 | 25.04 | 4.64 | 31.93 | 9.54 |
| ✓ | ✓ | ✓ | | 17.54 | 1.66 | 24.44 | 4.49 | 30.77 | 9.13 |

**Table 5: Study of the inputs and methods for strengthening underlying surface in USIR. If USSM is not used, MLP will be used instead.**

USIRs can significantly improve the denoising performance, particularly in point clouds with high noise levels. This trend suggests that USIRs can stabilize the underlying surface modeling process in noisy conditions. Run 2× testing phase denoising can achieve a small accuracy improvement, but need more computational resources. Notably, the 1× testing phase denoising with $\tau = 7$ can outperform the 2× testing phase denoising with $\tau = 3$, even though they undergo the same number of point position updates, as the latter re-extracts the underlying surface using the intermediate point cloud, while the former maintains a stable estimation of the underlying surface. The visualization in Fig. 7 shows that frequently re-extracting the underlying surfaces causes the points to converge towards a biased surface.

*The inputs and methods for strengthening underlying surface in USIR is studied in Tab. 5.* Without multi-scale neighborhood modeling for underlying surface representation strengthening using USSM, PD-Refiner degrades similarly to the fixed path version of PathNet, and the precision almost drops to that of the baseline model, demonstrating the significance of underlying surface representation strengthening. Both $F_l$ and $P_l^{disp}$ can promote the underlying surface estimation, but the effect of $P_l^{disp}$ is relatively minor due to its limited description of the underlying surface's location. When combining $F_l$ and $P_l^{disp}$, their complementary information provides a more effective depiction of the denoising process.

| #Points | | 10K Points | | | | |
|---|---|---|---|---|---|---|
| Noise Level | | 1% noise | | 2% noise | | 3% noise | |
| Loss type | | CD↓ | P2M↓ | CD↓ | P2M↓ | CD↓ | P2M↓ |
| $L2_{norm}$ | | 18.97 | 1.91 | 26.75 | 4.91 | 35.06 | 10.89 |
| CD | | 18.23 | 1.79 | 25.41 | 4.75 | 31.88 | 9.62 |
| $CD_{both-terms}$ | | 17.79 | 1.72 | 24.66 | 4.59 | 31.56 | 9.48 |
| $CD_{csw}$ | | 17.73 | 1.69 | 24.59 | 4.56 | 31.33 | 9.42 |
| $CD_{csw}$ + Adaptive | | 17.54 | 1.66 | 24.44 | 4.49 | 30.77 | 9.13 |

**Table 6: Study of stage adaptability and edge supervision strategies in ASPL. We use different loss type for supervision. $CD_{both-terms}$ is to add weight on both terms of CD loss.**

| Method | PD-Flow | Score | PSR | PathNet | IPFN | Ours$_{tiny}$ | Ours |
|---|---|---|---|---|---|---|---|
| Param (M) | 0.460 | 0.187 | 0.267 | 10.860 | 3.196 | 1.946 | 15.108 |
| FLOPs (G) | 4.508 | 21.281 | 102.284 | 2067.594 | 69.650 | 8.860 | 61.098 |
| Time (ms) | 72.2 | 152.7 | 220.6 | 2239.1 | 165.1 | 48.0 | 110.5 |

**Table 7: Compression of number of parameters (M), FLOPs (G) and inference time (ms).**

*The stage adaptability and edge supervision strategies in ASPL is studied in Tab. 6.* Compared to the standard Chamfer Distance (CD) loss, the edge-weighted CD loss achieves better performance. This accuracy is further improved when precise guidance is applied using the proposed clean-denoised shape-weighted CD loss. By providing hierarchical optimization targets for refiners in each layer, our method take the first place on denoising performance. We also use the $L2_{norm}$ loss proposed by IPFN and Score for training, achieving significant improvements compared to IPFN. This indicates that the underlying surface inheritance and strengthening paradigm is more efficient than stacking IterationModules.

*The training and testing computational resource requirements is studied in Tab. 7.* In processing 1K points patch with single 3090 GPU, we achieved higher efficiency. We also offer a tiny version with $C = 256$ and $\tau = 4$, which significantly improves efficiency with minimal accuracy loss. PathNet [57] needs to denoise each point in the point cloud independently, resulting in higher computational resource requirements. When training with 2× 3090 GPUs, PD-Refiner with $\tau = 8$ requires 34.57 GiB of training memory and 0.55s per iteration, whereas IPFN [5] with "4 it." requires 35.62 GiB and 0.52s per iteration. This results demonstrate PD-Refiner's relatively lower computational resource consumption.

## 5 Conclusion

In this paper, we propose a novel yet practical PD-Refiner for efficient point cloud denoising. Unlike previous iterative denoising methods, which replying on re-extract representations from intermediate point clouds, PD-Refiner inherits the underlying surface representation directly and leveraging USIRs to strengthen it by multi-scale neighborhood modeling. We also propose ASPL to provide USIRs with adaptive and more precise edges-aware supervision. Qualitative and quantitative results show that our method achieves excellent performance on both synthetic and real point cloud denoising benchmark.

Since we have found that the underlying surface representation inheritance and strengthening paradigm can make the denoising process more stable, we plan to extend our model to process data with more complex noise modalities, such as those from synthetic aperture radar and millimeter wave radar point clouds.

# Acknowledgments

This work is supported by the National Key Research and Development Program of China (Distributed Polarization 3D Imaging Radar System Technology, No. 2022YFB3901601).

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
