# OpenReview forum: "PD-Refiner: An Underlying Surface Inheritance Refiner with Adaptive Edge-Aware Supervision for Point Cloud Denoising"
_acmmm.org/ACMMM/2024/Conference — MM2024 Poster_

### Official Review · Reviewer_bKNq · 2024-05-09

**Rating:** 5
**Confidence:** 2

**Summary:**

Task: point cloud denoising


Contribution:
- a novel point cloud denoising framework, which extracts the underlying surface only once, thus boosting denoising stability
- ASPL (Adaptive Shape Preserving Loss) to ensure detailed and accurate edge restoration:  shallow USIRs have smaller edge weight; deeper USIRs have bigger edge weight

- SoTA performance while efficient

Method
- hierarchical USIRs: Underlying Surface inheritance Refiner: inherit and strengthen the point-wise underlying surface representation
    - two main submodules
        - PUM: Position Update Module (MLP-based) for point position updating: input l-th USIR's feature Fl, output its corresponding per-point position updates $P_l^{disp}$, used by $P_{l+1}^{m}=P_{l}^{m}+P_{l}^{disp}$
        - USSM: Underlying Surface Strengthen Module for feature updating (designed with MLP, PSA, etc.):  input $P_l^{disp}$ cat Fl, output feature updates: Ol, where $F_{l+1}=F_{l}+O_{l}$
    - Each USIR updates the features and point positions and finally obtained the output point positions

- Adaptive Shape Preserving Loss (ASPL):
    - get edge weights by SVWN: Shape Variance Weighting Network, input a clean pc, output per-point probability of belonging to an edge region
    - Add the edge weights for the  loss term of the nearest distance from clean to denoised point cloud
    - Deeper USIR with bigger edge weights


Summary:
This paper is overall well-written with good language. The novelty seems sufficient with reasonable and clearly stated motivation. The results are convincing with high performance and efficiency. My main confusion is on the implementation of SVWN. The description does not seem clear enough to understand.

**Strengths:**

- The novelty seems sufficient to me. The motivation is clear and reasonable to
    - avoid re-extracting the underlying surface by using the proposed USIR instead,
    - use multi-scale edge-aware loss to preserve geometry details
- Overall well-written with good language
- Experiments are convincing:
    - Performance seems good and convincing compared to SoTA methods including CVPR2023, both qualitatively and quantitatively.
    - Ablation study of the design of USIR and ASPL in Tables 4 and 5 are solid

**Limitations:**

Clarity can be improved:
- Details about SVWN: How is it trained to learn the edge regions? What loss is used? Is it supervised by GT? It's OK to explain in the supplementary file (which does not exist so far), but the current description is not sufficient.
- What does 'GAP' mean in Figure 4?
- What do different colors (mainly blue and yellow) mean in Figure 5 and Figure 7?
- Which dataset is Figure 7, Table 2? It's suggested to add the name of the dataset in the title of the figures or tables, like Table 1 and Figure 8.

presentation suggestion:
- Subsections are long. It's suggested to add subsubsections to make the paper easier to read. For example, in Subsection 3.3, divide the content into 'SVWN', 'ASPL'.
- Suggest adding qualitative results of the ablation study. It's ok to add in the supplementary file.

Other small questions:
- In Line 694, do you mean to say 'tested on two 3090 GPUs' instead of 'tested on 3090 GPUs'?
- In Table 4, why is ''USSM'' with ''w/o'' subscript ?

**Suitability:**

2

---

### Official Review · Reviewer_Yd2q · 2024-05-25

**Rating:** 4
**Confidence:** 3

**Summary:**

This work focuses on a well-studied problem, point cloud denoising. The core idea is based on some intuitive observation, thus proposing a practical point cloud denoising framework that boosts denoising stability by inheriting and strengthening the underlying surface representation.

**Strengths:**

1. The paper is well-motivated, and the organization is clear.
2. Figure 1 effectively explains the differences between the proposed method and existing methods at a high level.
3. This point cloud denoising paradigm is novel and interesting.
4. The experimental results outperform those of other methods.

**Limitations:**

1. In the abstract, it is not easy to understand the phrase "instability caused by transformations between the representation domain and the 3D space domain."
2. Some observations in Section 3 of the introduction are consistent with [1]. Therefore, I recommend that the authors add some discussions or experimental comparisons with [1].
[1] PathNet: Path-Selective Point Cloud Denoising
3. Additionally, some of the claims are too intuitive. The authors should add figures to support these claims. For example, the claim that "Subsequent iterations intensify these issues, as the re-extracted underlying surfaces lead to points converging towards a biased surface" should be well supported with visual evidence.
4. Regarding the experiments, solely providing some visualization evaluations on synthetic data is not sufficient. I encourage the authors to add quantitative comparisons on the real-scanned Kinect dataset [2].
[2] Mesh Denoising via Cascaded Normal Regression

**Suitability:**

2

---

### Official Review · Reviewer_zxzJ · 2024-06-04

**Rating:** 3
**Confidence:** 4

**Summary:**

The paper introduces PD-Refiner that inherits the underlying surface representation directly and leverages Underlying Surface Inheritance Refiner layers to strengthen it by multi-scale neighborhood modelling. They also propose the Adaptive Shape Preserving Loss to provide USIRs with adaptive training supervision across the different layers. Overall, they achieve promising results on both standard synthetic and real-world data.

**Strengths:**

The paper has several strengths:
- The paper is well-written and easy to follow.
- The results are promising, achieving state-of-the-art performance across most denoising settings.
- Overall, the method seems sound. However, there are some key issues that need to be addressed regarding the choice of loss function for training and also experimental results.

**Limitations:**

# Method and ablations
- In the introduction, the authors state that PD-Refiner is distinct from current iterative denoisers as current methods repeatedly iterate using the same encoder-decoder model. However, the PD-Refiner architecture has a lot of similarity with IPFN where the iterative denoising is conducted internally. The USIR layers (with the PUM and USSM) act similarly to the IterationModules of IPFN. I think the difference here is that the current method strengthens the underlying surface representation while IPFN re-extracts it from the updated points. Can you please clarify the distinction between the two a bit better as I believe both are methods that internally model denoising (as compared to PCN or Score).
- The loss ablation is only limited to Chamfer distance inspired loss functions. However, you use Chamfer distance as a metric in evaluation as well and I believe the choice of loss can have an impact on the evaluation results.
- By contrast, methods such as Score and IPFN minimize simpler L2 norm loss functions, between denoised points and the GT targets. Can you also provide ablation results when using such simpler loss formulations? This should demonstrate the method’s performance under the CD and P2M evaluation metrics without the reliance on a CD-inspired loss function. This way, we can evaluate if the proposed architectural changes have a significant improvement over Score and IPFN methods.
- The improvements made by CD_CSW and CD_CSW+Adaptive over CD_both-terms seems to be limited, and it seems that these loss functions have marginal improvements over the typical CD loss and the CD_both-terms loss. Can you please provide a discussion on why the improvement is so small?
- No supplementary code was provided to verify the implementation. I strongly suggest that the code is made accessible (if the paper is accepted), to ensure reproducibility.
# Experiments
- There is an issue with the P2M results for Table 1 (and perhaps with other results tables). Authors appear to have directly quoted the P2M results from each paper but I believe the P2M implementation settings are different in some papers compared to others (such as Score and IPFN using different settings). For example, in the IPFN original paper, their P2M results at 1% and 2% noise are lower than Score and PD-Flow, given their P2M implementation. Instead of directly quoting results, please re-evaluate the denoised outputs using a fixed P2M implementation with identical settings for all methods. Please provide the updated results, which should give a fair comparison.
- Table 2 only focuses on 10K points, can you please provide the comparison on point clouds with 50K points?
- In Rue Madame data, only 1 scene is provided. Can you provide additional results on other scenes? This can later be included as supplementary material.
- Also, can you provide results on Kinect data (see PCN and IPFN papers), which is also a real-world scan dataset, to show the generalization across Kinect noise pattern?
- Time per iteration and memory consumption are reported only for training. It is more important that we know the testing runtimes for the fully trained network versus SOTA methods, under their respective final hyperparameter settings.
- In addition to memory consumption, can you report the parameter numbers for your network, compared to other state of the art methods?
- When providing testing times, memory consumption and parameter number, versus other SOTA methods, please provide the results for the final implementation where the layer number is 7 to get a better understanding of the computational efficiency of this method w.r.t. others.
# Minor issues
- There are minor mistakes in the text such as line 805: “computational source”. Please carefully proofread.
- Line 479: 512/4 is 128 so I suggest not keeping the former as a fraction.

**Suitability:**

3

---

### Meta-Review · Area_Chair_zbto · 2024-06-27

**Recommendation:** Accept (Poster)
**Confidence:** 5

**Metareview:**

All reviewers vote for acceptance of this submission. Authors please follow reviewers comments carefully, e.g., the impact of different loss types, P2M consistency, additional comparisons, method details etc.